# Discrimination of Aluminum from Silicon by Electron Crystallography with the JUNGFRAU Detector

**Erik Fröjdh** [1,*][iD], **Julian T. C. Wennmacher** [2,3][iD], **Przemyslaw Rzepka** [2,3][iD], **Aldo Mozzanica** [1][iD], **Sophie Redford** [1][iD], **Bernd Schmitt** [1][iD], **Jeroen A. van Bokhoven** [2,3][iD] **and Tim Gruene** [4,*][iD]

1   Photon Science Division (PSD), Paul Scherrer Institut, 5232 Villigen, Switzerland; aldo.mozzanica@psi.ch (A.M.); sophie.redford@gmail.com (S.R.); bernd.schmitt@psi.ch (B.S.)
2   Energy and Environment Research Division (ENE), Paul Scherrer Institut, 5232 Villigen, Switzerland; julian.wennmacher@psi.ch (J.T.C.W.); przepka@ethz.ch (P.R.); jeroen.vanbokhoven@chem.ethz.ch (J.A.v.B.)
3   Institute for Chemical and Bioengineering, ETH Zurich, 8093 Zurich, Switzerland
4   Faculty of Chemistry, Department of Inorganic Chemistry, University of Vienna, 1090 Vienna, Austria
*   Correspondence: erik.frojdh@psi.ch (E.F.); tim.gruene@univie.ac.at (T.G.)

 

**Abstract:** The crystal structure of a chemical compound serves several purposes: its coordinates represent three-dimensional information about the connectivity between the atoms; it is the only technique that determines the absolute configuration of chiral molecules; it enables determining structure–function relations; and crystallographic data at atomic resolution distinguish between element types and serve as a confirmation of synthesis protocols. Here, we collected electron diffraction data from albite and from a Linde Type A (LTA) type zeolite. Both compounds are aluminosilicates with well-defined silicon and aluminum crystallographic sites. Data were recorded with the "adJUstiNg Gain detector FoR the Aramis User station" (JUNGFRAU detector) and we made use of its capability of energy discrimination to suppress noise. For both compounds, crystallographic refinement distinguishes correctly between silicon and aluminum, even though these elements have very similar electron scattering factors. These results highlight the quality of the electron diffraction data and the reliability of the models for chemical interpretation. Further development in this direction will provide enormous opportunities for structure–function studies by diffraction.

**Keywords:** charge-integrating detector; electron crystallography; aluminosilicates; discrimination of element types

## 1. Introduction

Hybrid pixel detectors (HPD) are widely used for X-ray crystallography both at synchrotrons and in-house set-ups. They are characterised by a high dynamic range, good linear response, and a neglectable read-out dead time. HPDs enabled shutterless continuous data collection and fine slicing of the rotation angle [1–3], and made it possible to adjust the data collection strategy to the scientific question and sample properties. The same characteristics make HPDs promising detectors in electron microscopes [4]. Previously, we demonstrated the qualities of the EIGER detector for electron diffraction [5]. Its technology was used in several other recent publications [6–9] that triggered a lot of attention [10–12].

In this work, we present how the "adJUstiNg Gain detector FoR the Aramis User station" (JUNGFRAU detector) [13] performs for electron diffraction experiments. In contrast to EIGER, and most other HPDs, JUNGFRAU is a charge-integrating detector instead of a single photon/electron counter. This is because it was designed for experiments at X-ray free electron lasers where all photons

arrive within femtoseconds, making it impossible to process pulses from individual particles. In-pixel gain switching provides a dynamic range of 120 MeV per pixel and frame, while offering single particle sensitivity down to energy depositions of 2 keV. Using a charge integrating detector has two distinct advantages over photon counting detectors: at low rates, where the tracks from single electrons are visible, the deposited energy per pixel can be used to estimate the entrance point [14] and at high rates the detector is not sensitive to pulse pileup. Like other hybrid pixel detectors, and in contrast to most monolithic active pixel sensors, the JUNGFRAU is radiation hard. Due to the sensor thickness, electrons do not penetrate to the pixel side, and thus cannot cause charge build up in the oxide layer and for bulk damage the electron would have to have at least 260 keV of energy [15]. Therefore, we expected, and have observed, no radiation damage to the detector from the electron beam intensities typically used for data collection.

We mounted a single JUNGFRAU module with a 320 μm thick silicon sensor onto a 200 keV transmission electron microscope with an $LaB_6$ electron source, and collected diffraction data from two aluminosilicate, the sodium feldspar mineral albite (sum formula $NaAlSi_3O_8$) and zeolite A (sum formula $NaAlSiO_4$) of Linde Type A (LTA) framework [16]. The framework of LTA is typically described in the space group of $Fm\bar{3}c$ (unit cell axis $a \approx 24.6$ Å) with two T-positions respectively assigned to silicon and aluminium (Si:Al = 1.0) without violation of Löwenstein's rule [17,18]. The unit cell is composed of eight large $\alpha$-cavities adjacent to one sodalite $\beta$-cage and d4r building units that results in a 3D topology. The electronegative framework of zeolite A is constructed by 8- 6- and 4-membered rings counterbalanced by 12 extra-framework $Na^+$ cations placed inside $\alpha$-cage. Sodium sitting in a plane of 8-ring controls the aperture size and determines the structure properties which have found broad application [19]. The framework of albite has four T-sites. All positions, including the $Na^+$ ion, are crystallographically ordered and fully occupied, and the sample is radiation hard. The spacegroup of albite is $P\bar{1}$. Zeolites are aluminosilicates and possess a crystalline, microporous structure. They are widely used as catalyst, ion exchanger and adsorbent. The framework consists of tetrahedrally-coordinated silicon and aluminum which occupy the so-called T-sites. These sites are connected by bridging oxygen atoms. The properties of zeolites are affected by the aluminum occupancy of the T-sites. In many catalytically active zeolites, the Si:Al ratio is high, and T-sites are only partially occupied with aluminum. One of the major remaining questions in the field of zeolites is the aluminum T-site occupancy. Both in albite and in zeolite A ($NaAlSiO_4$), the T-sites are fully occupied with either aluminium or silicon without mixed occupancies and with known ordering. Therefore, we chose these two compounds as control compounds for a project investigating zeolites with electron diffraction. For albite and zeolite A, we originally expected to differentiate between silicon and aluminum based on the differences in the T-O bond distance: they should be 1.61 Å for tetrahedrally coordinated Si-O and 1.76 Å for tetrahedally coordinated Al-O [20]. This work describes how we processed the diffraction data from the JUNGFRAU detector and presents the high quality data. As expected, our data sets mark the expected Al-position with an elongated T-O bond length compared with the T-O distances at the Si-sites. In addition, our diffraction data differentiate between $^{13}$Al and $^{14}$Si both during structure solution and after refinement of the structures, despite the small difference of the scattering power of these two close-by elements.

## 2. Materials and Methods

### 2.1. Instrumentation

Electron diffraction data were collected at 200 keV on a CM200 transmission electron microscope (TEM) (Philips, Amsterdam, Netherlands) with a $LaB_6$ electron source. The detector used was a $512 \times 1024$ pixel JUNGFRAU detector with a 320 μm thick silicon sensor and a pixel size of $(75 \text{ μm})^2$. Due to an aperture underneath the fluorescence screen, the detector area was restricted to a circle with about 320 pixels diameter. We therefore collected a low resolution data set at 780 mm effective detector distance, and a high resolution data set at 280 mm effective detector distance in order to cover the full resolution range of the samples. The CM200 has no C3 lens, which would enable Koehler illumination

for a parallel beam at a small beam diameter [21]. We set the beam diameter to match the instrument aperture at the chosen magnification. This may result in a non-parallel beam at the sample. However, this is not expected to have a serious impact on the data quality: XDS (see Section 2.4) reports a beam divergence of about $0.1°$. This value is of the same order of magnitude of what XDS reports when synchrotron X-ray radiation is used, and about an order of magnitude better compared with inhouse X-ray sources N.B.: The value reported as `BEAM_DIVERGENCE` by XDS is used to model the reflection profile, and not an exact measure for the true beam divergence.) The backfocal plane was focussed onto the detector plane by focussing the direct beam before every data collection. This procedure takes only a few seconds and could easily be automated by a TEM manufacturer. This is important for radiation sensitive samples. In order to facilitate this procedure even further, we implemented a live profile of the direct beam into the detector control interface, shown in Figure 1. During post processing, a threshold of 10 keV was applied to suppress electronic noise and low energy X-ray background. This value was chosen taking into account the measured noise of the detector (0.63 keV) in this configuration and allowing some margin for pedestal drift and low energy X-ray background.

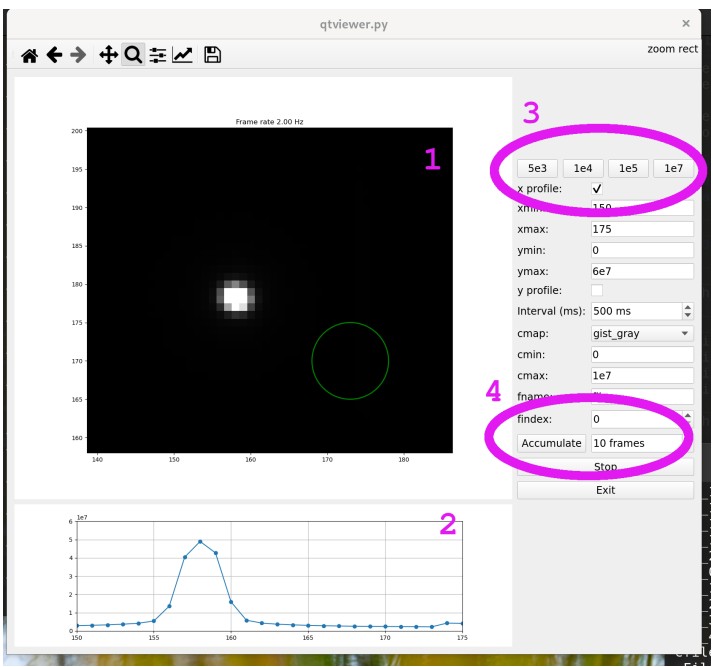

**Figure 1.** Graphical control interface of the JUNGFRAU detector developed for our experiments. (1) The main panel shows an enlarged image of the direct beam. The centre of the detector surface is shown as a green circle. (2) The one-dimensional beam profile below enables fast focussing of the diffraction pattern on the detector surface by minimising the standard deviation of the beam profile. The panel on the right sets the dimensions of the profile and the contrast of the main panel. (3) The top row has presets for the contrast. (4) The *Accumulate* button takes a screenshot.

## 2.2. Data Acquisition

Magnification in imaging mode was determined with a grating replica waffle, 2160 lines/mm (Ted Pella Product Number 607). The detector distance in diffraction mode and the direction of the rotation axis were estimated from the powder diffraction pattern of an evaporated aluminum standard (Ted Pella Product Number 619). Albite crystals were small. In their case, the rotation axis was centred with a beam diameter of 750 nm, at about 32,000 fold magnification. Zeolite A crystals were centred with a beam diameter between 2.3 µm and 3 µm, at 10,500 fold and 8,000 fold magnification. The same beam settings were used in diffraction mode. The first few data sets of zeolite A were collected with a cryo holder at $-189$ °C. Our goniometer stage, however, is optimised for the room temperature holder, and data collection was cumbersome because of drifts during rotation which could not be

corrected for. We therefore continued data collection with the room temperature holder. Prior to every data collection, the beam was manually focused on the JUNGFRAU detector by minimising the diameter of the direct beam. This was also monitored by a narrow profile of the direct beam (*cf.* Figure 1). The rotation rate at the CM200 is set manually with turn-button and varies with each data set. The rotation angle of the sample was recorded with a Digital Micrograph (DM) script [6], and the rotation rate was determined with a linear fit to the range of rotation [22]. The sample was always rotated in the positive direction from about $-40°$ to $+40°$. Rotation range at the aged CM200 is limited to this range, otherwise, one risks to break the vacuum. The order of steps during data collection are:

1. Start DM script to record rotation angle alpha
2. Start rotation
3. Start data capture from JUNGFRAU
4. Stop data capture from JUNGFRAU
5. Stop rotation

The duration of the DM script was set to 2–3 min, which was sufficient to cover the remaining steps. The rotation speed was aimed to be about $1°/s$ by setting it roughly to the same position for each data set.

To optimise the thermal stability of the detector, the JUNGFRAU was operated continuously at 1 kHz throughout the experiments. The detector was cooled with a Microcool M250 with Kryo 30 cooling liquid (Lauda-Brinkmann, LP, Delran, NJ, USA). The temperature was set to 14.5 °C. During searching and focusing, 100 frames at a time were summed and displayed in an online viewer, while data taking the stream was written to disk at 1 kHz. The exposure time was 980 µs to maximise the duty cycle. This mode of operation highlights the difference between single photon/electron detectors, which, due to their internal threshold, can be operated with long exposure times and charge integrating detectors which have to be run with a high frame rate due to leakage current. Pedestal measurements were taken directly after each measurement to reduce the impact of the middle and low gain pedestal collection procedure on the detector temperature.

*2.3. Data Conversion*

To convert the output of the detector, which consists of a 14 bit ADC value and 2 bits of gain data per pixels, we need three different calibration constants, one for each gain (G0, G1, G2), and three offsets for pedestal correction. The detector was calibrated using the standard procedure before mounting it on the microscope. The calibration consists of an absolute calibration of G0 with 8 keV photons from Copper X-ray fluorescence and relative calibrations for G1 and G2 using backplane pulsing and the internal current source [23].

Before processing with XDS, each frame was converted to deposited energy using the described procedure, then we applied a 10 keV threshold to suppress background before summing 50 frames together resulting in an effective frame rate of 20 Hz.

*2.4. Data Processing*

Data were processed with XDS. During indexing, rotation axis, unit cell constants, crystal orientation, and beam direction were refined (keyword `REFINE(IDXREF) = BEAM AXIS ORIENTATION CELL`), data were integrated with profile fitting, without parameter refinement (keyword `REFINE(INTEGRATE) = !`). During the scaling step (CORRECT in XDS), the same parameters as during indexing were refined for a first round (keyword `REFINE(IDXREF) = BEAM AXIS ORIENTATION CELL`). In subsequent rounds, the detector distance was refined instead of the beam direction (keyword `REFINE(IDXREF) = POSITION AXIS ORIENTATION CELL`). Simultaneous refinement of BEAM and POSITION resulted in unrealistic parameter drifts, most likely due to the high correlation between the beam direction, and the detector position [24]. Resolution cut-off was chosen approximately where $CC_{1/2}$ dropped below 70% [25] and $\langle I/\sigma_I \rangle$ dropped below 2.0. For albite, data from 9 crystals were

merged to reach near complete data (Tables S3 and S4). Because of the low symmetry spacegroup of albite, $P\bar{1}$, scaling of individual datasets was suppressed, and all datasets together were scaled with default options in XSCALE [26,27].

Both the detector distance and the rotation axis could be refined during data processing. High resolution data of our samples most likely stabilised the refinement of the detector distance and may not be possible for samples that diffract to less than 0.9'ish Å resolution.

## 2.5. Structure Solution and Refinement

Structures were solved with SHELXT [28]. SHELXT makes use of normalized structure factors. These correct for the $\theta$-dependent fall-off of the structure factor, making it independent from the type of radiation [29]. The structure of albite was found from the default options. For the structure of zeolite A, we enforced the known spacegroup $Fm\bar{3}c$ with the SHELXT command line switch '-s"Fm-3c"', as otherwise the non-centrosymmetric spacegroup $F\bar{4}3c$ was chosen. This is most likely due to the more stringent requirements for centrosymmetric spacegroups [30]. Structure was built with SHELXLE [31] and refined with SHELXL [32]. Scattering factors tabulated in Table 4.3.1.1 of [33] were fitted with the Cromer–Mann-parametrisation available in SHELXL. Starting values for the fitting were taken from the parametrisation in [34]. The nine parameters of the Cromer–Mann-parametrisation were fitted with GNUPLOT [22]. Only neutral scattering factors were used, no ionic scattering factors. All atom coordinates were refined independently and without restraints. Atomic displacement parameters (ADP) were refined isotropically. Anisotropic ADPs can act as fudge factors that would wash out the small differences between the T1/T2 configurations. A few hundred cycles of least-squares refinement were typically sufficient to reduce the parameter shifts to zero. This "zero-shifts" state was reached for every configuration before reporting the R1 and $R_{complete}$ values listed in Tables 1 and 2, as well as Figures 2 and 3. In the structure of zeolite A, the strongest sodium atom, at the 6-ring, was typically assigned as oxygen atom by SHELXT. The second sodium, at the 8-ring, visible as a peak in the difference map and placed accordingly. We did not model the third sodium ion because of its very low occupancy of about 4 %, *cf.* [17].

**Table 1.** R1 and $R_{complete}$ [35] values of albite for all 16 combinations of silicon and aluminum across the four T-sites of albite. The first column lists the number of T-sites modelled as aluminum, the following four columns list which T-sites were modelled as aluminum and silicon respectively. Atom coordinates and one isotropic ADP value were refined for each atom site without restraints, and until convergence was reached (zero parameter drift reported by SHELXL). Values in parentheses for strong reflections ($I > 2\sigma(I)$).

| # Al | T1 | T2 | T3 | T4 | R1 | $R_{complete}$ |
|---|---|---|---|---|---|---|
| $0 \times$ Al | Si | Si | Si | Si | 22.80 (17.82) | 23.11 (19.78) |
| $1 \times$ Al | Si | Si | Si | Al | 22.73 (17.76) | 23.05 (19.74) |
| | Si | Si | Al | Si | 23.08 (18.11) | 23.41 (20.07) |
| | Si | Al | Si | Si | 23.06 (18.08) | 23.37 (20.12) |
| | Al | Si | Si | Si | 23.05 (18.10) | 23.38 (20.06) |
| $2 \times$ Al | Si | Si | Al | Al | 22.93 (17.98) | 23.25 (19.95) |
| | Si | Al | Si | Al | 22.96 (17.95) | 23.25 (19.97) |
| | Al | Si | Si | Al | 22.95 (17.98) | 23.29 (19.98) |
| | Si | Al | Al | Si | 23.26 (18.29) | 23.60 (20.31) |
| | Al | Si | Al | Si | 23.26 (18.31) | 23.59 (20.26) |
| | Al | Al | Si | Si | 23.24 (18.27) | 23.56 (20.31) |
| $3 \times$ Al | Si | Al | Al | Al | 23.08 (18.13) | 23.41 (20.14) |
| | Al | Si | Al | Al | 23.14 (18.20) | 23.48 (20.16) |
| | Al | Al | Si | Al | 23.10 (18.09) | 23.43 (20.15) |
| | Al | Al | Al | Si | 23.41 (18.42) | 23.74 (20.45) |
| $4 \times$ Al | Al | Al | Al | Al | 23.18 (18.22) | 23.50 (20.22) |

**Table 2.** R1 and $R_{complete}$ [35] values of zeolite A, measured at room temperature and $-189\,^{\circ}$C (as indicated). The second T-site is the aluminum site, i.e., the first column contains the correct combination Si/Al. All four possible combinations of modelling the two T sites as silicon and aluminum, respectively. Atom coordinates and one isotropic ADP value were refined for each atom site without restraints, and until convergence was reached (no parameter drift). Number in brackets calculated from strong reflections $(I/\sigma_I > 2)$.

| Sample | | T1 Modelled as/T2 Modelled as | | | |
|---|---|---|---|---|---|
| | | **Si/Al** | **Si/Si** | **Al/Al** | **Al/Si** |
| x1 (RT) | R1 [%] | 32.62 (31.42) | 32.67 (31.32) | 33.83 (32.51) | 33.95 (32.50) |
| | $R_{complete}$ [%] | 33.51 (32.31) | 33.54 (32.19) | 34.94 (33.62) | 34.96 (33.50) |
| x2 (RT) | R1 [%] | 30.32 (29.75) | 30.66 (30.11) | 31.23 (30.69) | 31.18 (30.55) |
| | $R_{complete}$ [%] | 31.18 (30.61) | 31.47 (30.92) | 32.22 (31.68) | 32.07 (31.45) |
| x3 ($-189\,^{\circ}$C) | R1 [%] | 27.47 (24.15) | 27.67 (24.19) | 28.01 (24.42) | 28.10 (24.65) |
| | $R_{complete}$ [%] | 29.12 (25.82) | 29.30 (25.82) | 29.70 (26.10) | 29.87 (26.43) |
| x4 ($-189\,^{\circ}$C) | R1 [%] | 30.80 (28.39) | 30.87 (28.36) | 31.29 (28.61) | 31.26 (28.80) |
| | $R_{complete}$ [%] | 31.82 (29.37) | 31.83 (29.31) | 32.35 (29.65) | 32.35 (29.88) |
| x5 ($-189\,^{\circ}$C) | R1 [%] | 27.42 (26.76) | 27.57 (26.92) | 27.99 (27.25) | 28.28 (27.65) |
| | $R_{complete}$ [%] | 28.30 (27.65) | 28.44 (27.79) | 28.93 (28.19) | 29.25 (28.63) |

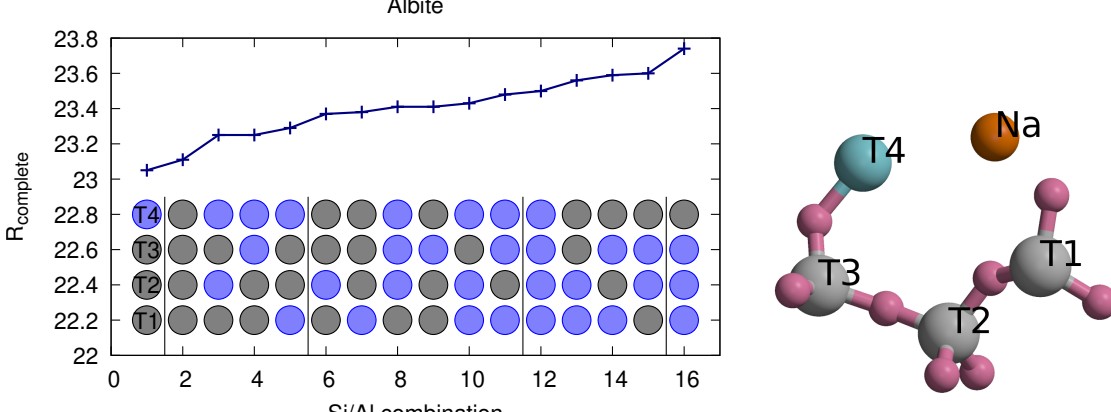

**Figure 2.** Left: $R_{complete}$ for all 16 combinations for assigning aluminum (blue circle) and silicon (grey circle) to the four T-sites in albite, sorted increasingly. The lowest $R_{complete}$ corresponds to the correct combination $T1 = T2 = T3 = Si$ and $T4 = Al$. $R_{complete}$ increases with the number of wrongly assigned element types (grouped by vertical lines). Atom coordinates and one isotropic ADP value were refined for each atom site without restraints, and until convergence was reached (no parameter drift). Right: Asymmetric unit of albite with labelling of the four T-sites. T4 corresponds to aluminium. Oxygen atoms red, the sodium ion in orange.

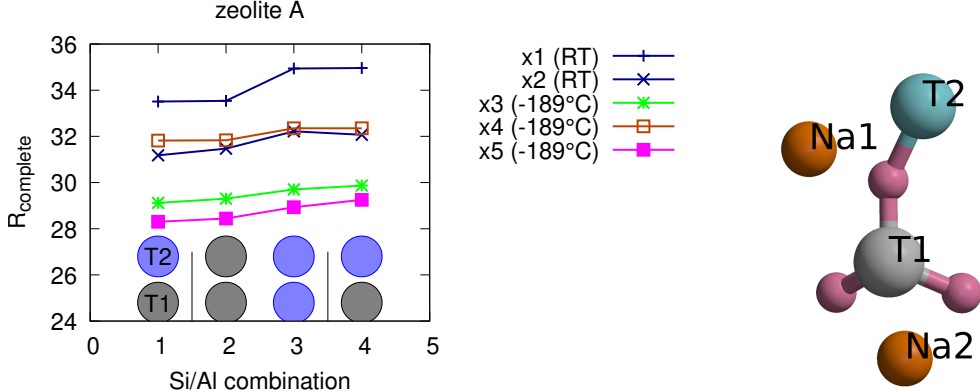

**Figure 3.** Right: $R_{complete}$ for all four possibilities to model T1 and T2 as aluminum (blue circle) and silicon (grey circle), respectively, plotted for all five crystals. The lowest $R_{complete}$ corresponds to the correct combination T1 = Si and T2 = Al. Atom coordinates and one isotropic ADP value were refined for each atom site without restraints, and until convergence was reached (no parameter drift). Left: Asymmetric unit of zeolite A with its two T-sites and two sodium ions (orange). We did not model the third, weakly occupied sodium ion [17].

*2.6. Data Availability*

The diffraction data in CBF format, together with the input files for XDS, are available from zenodo.org, DOI 10.5281/zenodo.4249684. The metadata of the CBF files only contain the pixel size and the wavelength. The full geometric description of the experiments can be found in the accompanying XDS files. Structural data have been deposited at the ICSD with the codes

| Albite | CSD 2042879 | LTA x1 | CSD 2042880 | LTA x2 | CSD 2042881 |
|--------|-------------|--------|-------------|--------|-------------|
| LTA x3 | CSD 2042882 | LTA x4 | CSD 2042883 | LTA x5 | CSD 2042884 |

## 3. Results

*3.1. Operation of the JUNGFRAU Detector and Data Conversion*

The JUNGFRAU detector was not integrated into the control electronics of the transmission electron microscope (TEM), meaning that we had to resort to manual data collection. The graphical user interface (GUI), shown in Figure 1, facilitates this process as much as possible: It provides a live view suitable for sample search in imaging mode of the microscope, and during data collection of the diffraction data. The one-dimensional profile at the bottom of the GUI helps focus the beam and thus the diffraction pattern on the detector surface. This greatly relaxes the requirement of a parallel beam that is often time-consuming to adjust, and enables the use of a beam diameter smaller than what would be possible with a strictly parallel beam [21]. The standard deviation of the beam profile is narrowest when the beam is focussed. Screenshots can be recorded directly from the GUI both in imaging and diffraction modes of the TEM. The live-view during data collection provides the trained crystallographer with a first impression of the data quality, and data collection can be stopped, e.g., when the crystal moves out of the beam, or loses diffraction power due to radiation damage.

The JUNGFRAU detector has three different gains. Each pixel switches to the next gain when the accumulated charge reaches a specific threshold. The pixel encodes its gain setting (Figures 4b and 5b) together with analog data, which can then be converted to deposited energy from a detector specific gain map. We recorded the pedestal in each of the three gains after every data collection (Figures 4c and 5c). After conversion, the data per pixel corresponds to the energy deposited during the 1 ms exposure time (Figures 4d and 5d). Due to the low intensity impinging onto the detector, the diffraction pattern becomes visible to the bare eye only after summation of 50 frames, i.e., 50 ms of exposure per frame (Figures 4e and 5e). During data conversion, a threshold of

10 keV was applied for noise suppression. The effect is illustrated in Figure 6. It shows a data frame of zeolite A with (left) and without (middle) application of the 10 keV threshold. The difference between both frames is not obvious. However, substracting the non-thresholded frame from the thresholded frame reveals additional reflections, which are invisible on the original frames (right).

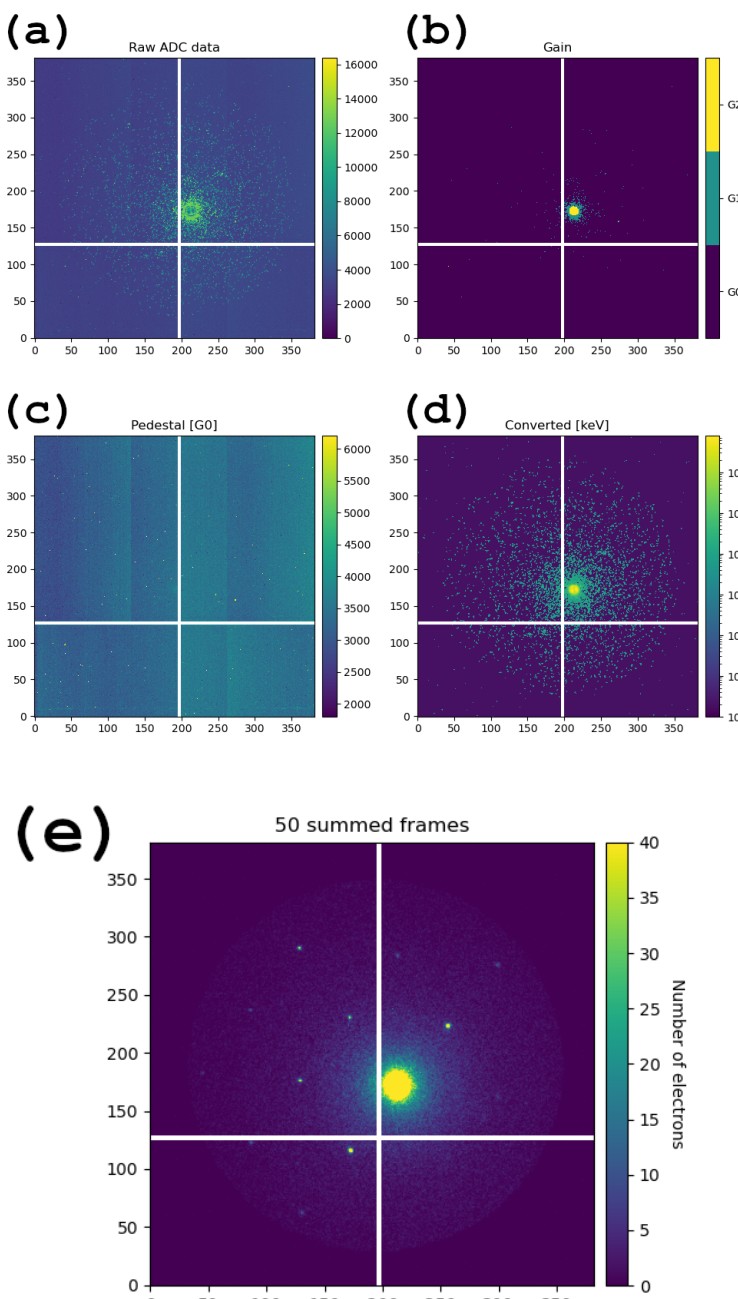

**Figure 4.** Albite data conversion and energy correction with the JUNGFRAU detector. (**a**) Raw data recorded in 1 ms, (**b**) gain levels per pixel, (**c**) pedestal map for gain level 0, (**d**) converted frame, (**e**) sum of 50 converted frames makes the diffraction pattern visible.

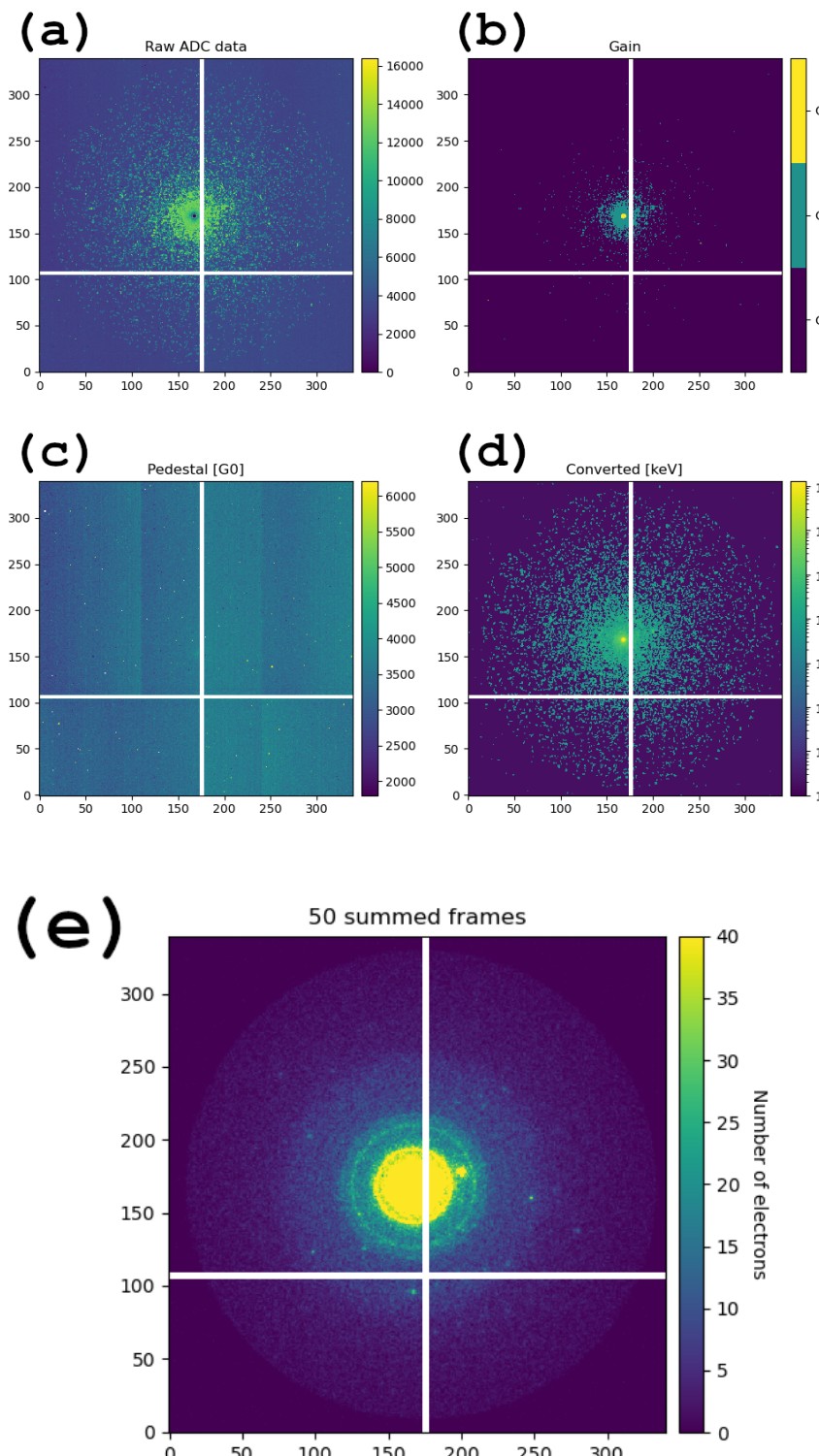

**Figure 5.** Similar to Figure 4, zeolite A data conversion and energy correction with the JUNGFRAU detector. (**a**) Raw data recorded in 1 ms, (**b**) gain levels per pixel, (**c**) pedestal map for gain level 0, (**d**) converted frame, (**e**) sum of 50 converted frames makes the diffraction pattern visible.

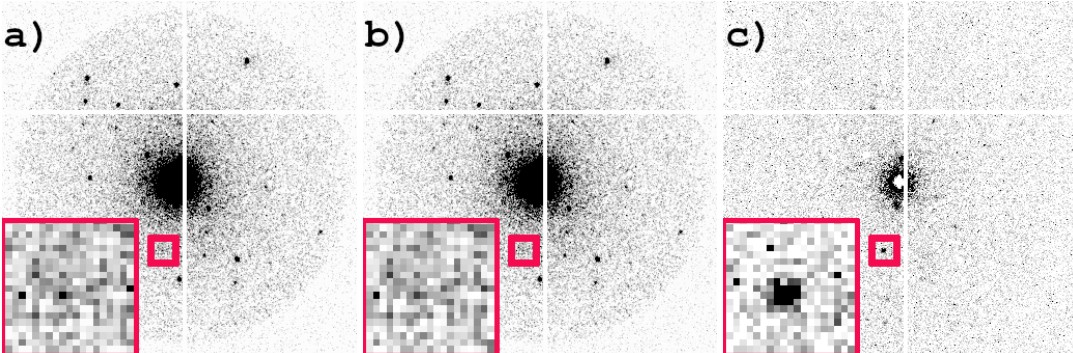

**Figure 6.** Noise reduction by application of a 10 keV threshold. (**a**) Frame with 10 keV threshold applied, (**b**) frame without threshold applied, (**c**) frame (**a**)–frame (**b**) reveal "hidden" reflections. The insets show a magnified version of the region of interest. Images including background subtraction with ADXV [36].

### 3.2. Benchmark Sample Albite

Albite has four T-sites. Therefore, there are 16 possible combinations to model each T-site as either silicon or aluminum. The crystallographic residual values R1 ($R1 = \sum_{hkl} \frac{||F_{\text{obs}}(hkl)|-|F_{\text{calc}}(hkl)||}{|F_{\text{obs}}(hkl)|}$) for each combination are listed in Table 1. The known correct combination, $T1 = T2 = T3 = Si$ and $T4 = Al$, results in the lowest R1 and $R_{\text{complete}}$ ($R_{\text{complete}}$ reduces the effect of overfitting and is more sensitive to small differences in the structural model than R1 [35]). Figure 2 plots these values sorted by $R_{\text{complete}}$. The vertical bars group combinations by the number of wrong element types, e.g., the correct model has zero wrongly modelled T-sites, while the second model, with all T-sites modelled as silicon, has one T-site modelled wrongly, namely T4. This graph supports our claim that the lowest $R_{\text{complete}}$ for the correct model is due to data quality rather than pure coincidence, as the $R_{\text{complete}}$ rises with an increasing number of wrongly modelled T-sites (up to $T1 = T2 = T3 = Al$ and $T4 = Si$, where each T-site is modelled with the wrong element). Observed data, calculated data, and their difference are plotted for each of the 16 possible combinations in Supplemental Figures S1–S3. The differences between each combination is hardly visible by the bare eye, which reflects the small differences in the residual values.

Originally, the high quality of our data became apparent from the structure solution even before refinement. The asymmetric unit of albite has the sum formula $Si_3AlO_8Na$. The structure solution with SHELXT correctly assigned all atoms, except for the $Na^+$ ion, which was assigned as an additional oxygen atom. Three T-sites were assigned as silicon, one element was assigned as aluminum, all at their expected positions. SHELXT decides about the element type from the summed density values within a 1.4 Å sphere around map peaks and assigns the strongest peak(s) to the heaviest atom from the list of elements provided by the user ($O, Na, Si, Al$ in this case of albite). The remaining element types are assigned based on the relative peak strengths according to the list of elements [28]. This means that the map value at the aluminum T-site differs sufficiently from the three silicon T-sites already before model refinement, despite seemingly poor data statistics (*cf.* Table S3). Note that the sodium atom is not bonded and has higher ADP values than the framework atoms. Consequently, the local map value is decreased, which is most likely the reason why it was incorrectly assigned as oxygen. We could confirm the correctness of the assignment from the T-O bonds. The expected Al-O bond is about 1.76 Å and the Si-O bond is about 1.61 Å [20]. The Al-O and Si-O bond lengths for our model are 1.71–1.75 Å and 1.58–1.64 Å, respectively (Table S1).

### 3.3. Benchmark Sample Zeolite A

Zeolite A has only two T-sites and four possible combinations to model each T-site as either silicon or aluminum. Zeolite A has spacegroup $Fm\bar{3}c$. This high symmetry space group results in complete data

from individual crystals, and we could confirm our findings by comparing results from several crystals. The crystallographic residual values for five different crystals are tabulated in Table 2 and plotted in Figure 3. Observed data for zeolite A are compared with the calculated data in Supplemental Figure S5. As with albite, SHELXT assigned all element types correctly, except for sodium, i.e., it identified the correct T-site for aluminum and the remaining three for silicon. The average T-O bond lengths confirmed the correct assignment (see exemplary Table S2). In every case, the model with the correct assignment results in the lowest $R1/R_{complete}$-values, listed in Table 2 and illustrated in Figure 3.

## 4. Discussion

We collected electron diffraction data at 200 keV with a JUNGFRAU detector and suppressed noise below 10 keV using its capability for energy discrimination in the keV range, intrinsic of the detector itself. We merged low and high resolution data to ensure complete data across the full resolution range, including low resolution data. The detector does not require a beam-stop, and we focussed the beam in the detector plane in diffraction mode. This relaxes the requirements for a parallel beam and facilitates data collection. We demonstrate the high quality of our data with several data sets from albite and zeolite A, an LTA type framework. Our data reliably distinguishes aluminum from silicon based on their scattering factors both during structure solution, and for the refined model. While a 200 keV TEM with a $LaB_6$ electron source is very well suited for electron diffraction studies [37], many TEMs are equipped with a field emission gun and are usually operated at 300 keV. For operation at 300 keV, one would choose a thicker silicon layer, e.g., 450 μm, to shield the ASIC. Depending on the dose requirements, one would look into specialized sensor designs [38]. Several experiments have been performed successfully at 300 keV, showing that the current generation of detectors is useful at this energy, but, to the knowledge of the authors, no systematic measurements of radiation hardness has been carried out. Hybrid Pixel Detectors for Electron Microscopy should also benefit from the vast amount of research going into radiation hard sensors and ASICs for the High Luminosity LHC.

The silicon and aluminum positions of our benchmark samples, albite and zeolite A are well known, and they have full occupancy, i.e., there is no disorder between silicon and aluminum. Our findings raise the question as to what extent electron diffraction could differentiate disordered structures, where one T-site is occupied both by aluminum and silicon. This question is important for understanding the chemical properties of zeolites and to tune their catalytic properties. We are convinced that further method development for electron diffraction will shed light on this question. Here, we refrain to provide a quantitative estimate of the occupancy level that electron diffraction will eventually differentiate. There are too many factors that currently affect the data quality and in particular the noise level, which shadows the signal difference between aluminum and silicon. These factors are of both and an instrumental and methodical nature: Electron diffraction is almost always affected by dynamic scattering, while both scaling and refinement assumes kinetic scattering. Furthermore, scaling does improve data quality, and the solvability for structures with *ab initio* methods. However, the crystallographic scaling methods of programs like XDS, AIMLESS, or SADABS are tuned for X-ray diffraction [39–42]. A scaling algorithm tuned for electron diffraction still needs to be developed. Data collected with the rotation method assume a constant oscillation width, i.e., a reliable goniometer. This may not be guaranteed, in particular with old instruments like we have access to. Finally, scattering factors can most likely be improved, and we made no attempt to use ionic scattering factors [43]. More sophisticated electron scattering factors than those from [33] are available [43,44]. Furthermore, we processed our data with XDS. Other data reduction programs, like PETS, can be more easily combined with dynamical refinement, e.g., with JANA2006, which can reveal finer structural details than a pure kinematic approach [45–48].

## 5. Conclusions

The JUNGFRAU detector is a charge-integrating hybrid pixel detector. Due to its in-pixel gain switching, it has a dynamic range of 120 MeV, and it can discriminate particle energies in the keV range.

We demonstrate here that the JUNGFRAU detector is suitable for recording electron diffraction data. We developed a graphical user interface for convenient data collection. We collected crystallographic diffraction data of excellent quality: the data are good enough to reveal the subtle difference between aluminum and silicon in two different types of aluminosilicates. We are confident that we can further improve the data quality. In the context of zeolite research, we aim at the determination of the occupancy level of T-sites with mixed aluminum and silicon occupancies. One of the possible improvements exploits the fast read-out rate of the JUNGFRAU detector of up to 2 kHz. Combined with low dose exposure, this enables particle tracking of the impacting electrons. This will improve data in two ways: knowing the point of impact will improve the intensity and the background estimates for every reflection, and the multiple counting of single electrons [5] will be avoided.

**Supplementary Materials:** The following are available online at http://www.mdpi.com/2073-4352/10/12/1148/s1, Figure S1: Data for albite structure depending on the T1–T4 compositions, Figure S2: S1 Cont'd: Data for albite structure depending on the T1–T4 compositions, Figure S3: S1 Cont'd: Data for albite structure depending on the T1–T4 compositions, Figure S5: Data for Zeolite A structure depending on the T1/T2 composition, Table S1: T-O bond lengths for the refined structure of albite, Table S2: T-O bond lengths for the refined structure of zeolite A, crystal 3, Table S3: Information for data sets from 9 crystals merged for albite, Table S4: Data statistics for albite after merging of data sets from 9 crystals.

**Author Contributions:** Conceptualization, T.G., J.T.C.W., and E.F.; software, E.F.; detector preparation and calibration, A.M., S.R.; validation, J.T.C.W., T.G.; formal analysis, T.G. and J.T.C.W.; resources, J.A.v.B., B.S., T.G.; data curation, E.F., T.G., P.R.; writing—original draft preparation, E.F., T.G., J.T.C.W., P.R.; writing—review and editing, all; funding acquisition, B.S., J.A.v.B., T.G. All authors have read and agreed to the published version of the manuscript.

**Funding:** This research was funded by the Swiss National Science Foundation, Grant No. 200021_169258.

**Conflicts of Interest:** The authors declare no conflict of interest. The funders had no role in the design of the study; in the collection, analyses, or interpretation of data; in the writing of the manuscript, or in the decision to publish the results.

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
