# Peer review of "Discrimination of Aluminum from Silicon by Electron Crystallography with the JUNGFRAU Detector"

_crystals, doi:10.3390/cryst10121148_

Round 1

Reviewer 1 Report

This manuscript presents a new hybrid pixel detector which uses charge integrating detection rather than the single counting mode which have been part of the recent revolution in electron detection in the TEM field. These detectors have the advantage of being able to energy discriminate the electrons and detect many events simultaneously. These detectors also have enormous dynamical range compared to the the current generation of detectors. The authors claim the data quality collected with this detector improve the analysis to the point where the software can automatically assign elements in the structure which are close in the periodic table. 

The overall structure of the paper is well thought out and implemented, I do however have some questions, and comments. 

You suggest that 200KeV I step typical energy used for 3ED data collection, there are a significant number of TEM which run and collect 3DED data at 300KeV which may cause damage to this detector, which you state has a damage threshold of 260KeV. I would like yo auto reword this section of the paragraph to reflect the fact that there are a significant number of TEMs which are above this damage threshold, this higher accelerating voltage is often desirable because it allows thicker samples to be collected and still have limited dynamical scattering within the data. 

You state “data differentiate between 13Al and 14Si both during structure solution with direct methods, and after refinement of the structures” this is indeed a big step forward for 3DED data however this is no mention of which scattering factors are used or the role they may have played in the direct methods and refinement results, does changing the scattering factors change the outcome? There are several sets of electron scattering factors available, please let th reader know which ones you are using and the impact of using different scattering factors will have eon this result. 

An additional factor which maybe in your favour when comparing the correct assignment of Silicon and Aluminium is that the cross over of the scattering factor occurs at low resolution, where as for elements which are further apart the crossover in the scattering factors occurs at medium resolution, which often confuses the assignment algorithms in software like SHELX or SIR. Is it possible to compare with a crystal where the elements are close but the crossover in scattering factor occurs at higher scattering angle? This way we can ascertain if this is the improved data quality or the scattering factors which are leading to this note worthy and impressive result of distinguishing between adjacent elements with electron scattering data. 

“Pedestal measurements were taken directly after each measurement to reduce the impact of the middle and low gain pedestal collection procedure on the detector temperature. “ If his detector was to be used as a detector for 3DED data collection would it be necessary to collect the pedestal measurements along with each data set? How do you generate the 8KeV X-rays in the TEM, is it as simple as using the 200KeV beam to onto copper? Please clarify for the reader. 

For table S3 can you explain how the data was merged, what is the contribution form each crystal to the data set. How do the crystals vary in morphology and size which may impact the amount dynamical scattering present in the datasets, is this accounted for, if not, why not, please explain to the reader. What potential impact does this have on the discrimination of the Al and Si? Would merging different aspects of the data lead to different results? 

Given the small difference between the R values and the refinement is kinematic on electron data which is highly likely to be have some dynamical content to the data, can you compare the observed data to theoretical data in resolution shells and determine the dynamical content of the high resolution data which would be the most discriminatory between the Al and Si positions, then comment for the reader the effects of the dynamical content of the high resolution data and how this may impact upon the result. 

In the supplementary section S3 why are some uncertainties limited to the lower resolution shell of 6.36 to 2.86?

Author Response

Comments and Suggestions for Authors

This manuscript presents a new hybrid pixel detector which uses charge
integrating detection rather than the single counting mode which have been part
of the recent revolution in electron detection in the TEM field. These detectors
have the advantage of being able to energy discriminate the electrons and detect
many events simultaneously. These detectors also have enormous dynamical range
compared to the the current generation of detectors. The authors claim the data
quality collected with this detector improve the analysis to the point where the
software can automatically assign elements in the structure which are close in
the periodic table.

The overall structure of the paper is well thought out and implemented, I do
however have some questions, and comments.

You suggest that 200KeV I step typical energy used for 3ED data collection,
there are a significant number of TEM which run and collect 3DED data at 300KeV
which may cause damage to this detector, which you state has a damage threshold
of 260KeV. I would like yo auto reword this section of the paragraph to reflect
the fact that there are a significant number of TEMs which are above this damage
threshold, this higher accelerating voltage is often desirable because it allows
thicker samples to be collected and still have limited dynamical scattering
within the data.
############################################################################
# response: We added a section to the Discussion (" While a 200~keV TEM with a
# $LaB_6$ electron source is very well suited for electron diffraction studies
# \cite{a3edpi_design:2019}, many TEMs equipped with a field emission gun are
# usually operated at 300~keV. For operation at 300 keV one would choose a
# thicker silicon layer [...]")
############################################################################

You state â<80><9c>data differentiate between 13Al and 14Si both during structure
solution with direct methods, and after refinement of the structures<80><9d> this is
indeed a big step forward for 3DED data however this is no mention of which
scattering factors are used or the role they may have played in the direct
methods and refinement results, does changing the scattering factors change the
outcome? There are several sets of electron scattering factors available, please
let th reader know which ones you are using and the impact of using different
scattering factors will have eon this result.

######################################################################
#response:
- section 'Structure Solution and Refinement;' added sentence about normalised
structure factors, and reference to Giacovazzo's textbook.
- also here: added description of scattering factors, with reference to Int.
Tables Vol. C, and Peng (1999)
- section 'Results': added sentence w.r.t. P. Dominiak's work on improved
scattering factors and Palatinus' dynamical refinement
############################################################################

An additional factor which maybe in your favour when comparing the correct
assignment of Silicon and Aluminium is that the cross over of the scattering
factor occurs at low resolution, where as for elements which are further apart
the crossover in the scattering factors occurs at medium resolution, which often
confuses the assignment algorithms in software like SHELX or SIR. Is it possible
to compare with a crystal where the elements are close but the crossover in
scattering factor occurs at higher scattering angle? This way we can ascertain
if this is the improved data quality or the scattering factors which are leading
to this note worthy and impressive result of distinguishing between adjacent
elements with electron scattering data.

############################################################################
#response: SHELXT uses normalised structure factors, and does not take the type
of radiation into account (also cf. response reviewer 2). Several people
reported the discrimination of C/N/O, e.g. the paracetamol or methylene blue
structure in Gruene et al, Angew. Chemie (2018), although we are not aware of a
qualitative investigation with emphasis on reproducibility. We looked into
several data sets (for zeolite A) and two different samples, and checked every
possible T-site. The relative differences N:C or O:C are larger than Si:Al. As
we also explain in response to reviewer 2, we would like to present our results
as information, and not as ranking. We were impressed with our results, which is
why we made them topic for this manuscript, but the main objective is a report
of how we use the JUNGFRAU detector for our ED studies, not to claim it is
better than other detectors. We believe that the manuscript reflects our
attitude.
############################################################################

"Pedestal measurements were taken directly after each measurement to reduce the
impact of the middle and low gain pedestal collection procedure on the detector
temperature." If his detector was to be used as a detector for 3DED data
collection would it be necessary to collect the pedestal measurements along with
each data set?

############################################################################
#response: In short: no, one would not need to redo pedestal before each dataset,
# the longer answer is that it depends on the temperature stability of the
# setup, exposure time and if there is any radiation damage. With a very good
# chiller and short integration times one could probably live with the same
# pedestal for weeks...
############################################################################

How do you generate the 8KeV X-rays in the TEM, is it as simple
as using the 200KeV beam to onto copper? Please clarify for the reader.

######################################################################
#response: clarified that this was done only once, prior to mounting the
detector in the microscope. we added in section "Data Processing" '[...] the
standard procedure before mounting in the microscope. The calibration consists
of an absolute calibration [...]"
############################################################################

For table S3 can you explain how the data was merged, what is the contribution
form each crystal to the data set. How do the crystals vary in morphology and
size which may impact the amount dynamical scattering present in the datasets,
is this accounted for, if not, why not, please explain to the reader. What
potential impact does this have on the discrimination of the Al and Si? Would
merging different aspects of the data lead to different results?

############################################################################
#response: the data sets were merge in XSCALE with default options. We inserted
a new Table S3 with some information about individual data sets and added a
sentence "Because of the low symmetry spacegroup of albite $P\bar{1}$, scaling was
suppressed during data integration, and datasets were scaled with default
options in XSCALE. to section 'Data processing'.
Other than absorption correction applied by XDS and XSCALE, we did not account
for crystal morphology or size. Table S3 lists the scaling factors for the
individual crystals. The crystals were small, but except for one
crystal, we did not record images. We took no special care w.r.t. dynamic
scattering. The theory of dynamic scattering, as far as we understand, is
optimised for crystals in specific crystallographic orientations, while the
strength of the rotation method lies in arbitrary orientation of the crystal.
This averages out many errors. Dynamic scattering is among these errors, but
also other errors. Lukas Palatinus' program JANA2006 can carry out dynamic
refinement. But last time we discussed with Lukas Palatinus, it is work in
progress to make dynamic refinement compatible with XDS integrated data. We
added this statement at the end of the Results section. We are looking into
improved scaling algorithms for ED, but is not part of this manuscript.
############################################################################

Given the small difference between the R values and the refinement is kinematic
on electron data which is highly likely to be have some dynamical content to the
data, can you compare the observed data to theoretical data in resolution shells
and determine the dynamical content of the high resolution data which would be
the most discriminatory between the Al and Si positions, then comment for the
reader the effects of the dynamical content of the high resolution data and how
this may impact upon the result.
############################################################################
# response: we are not convinced that dynamic scattering has the effect
# described by the reviewer. E.g. for zeolite A, x5 (ltacryo_manu_x3_x.cif):

  • SHEL 1 0: the lowest R1-value is no longer the correct model. Instead:
    - SHEL 4 1: (suggested by JAvB, where the difference between Si and Al is
    strongest w.r.t kinematic scattering (cf. Fig. 7 in original
    submission),correct order is even more than with all data.
    Programs like XDS scale diffraction data. As far as we understand from
    discussion with developers of ED with a background in materials science
    (rather than e.g. X-ray protein crystallography), scaling is not a well known
    concept in the field. We assume that scaling as a strong influence on making
    the intensities more kinematic.
    We decided not to add this discussion to the manuscript, as we believe our
    understanding of the phenomena is not profound enough.
    ############################################################################

In the supplementary section S3 why are some uncertainties limited to the lower
resolution shell of 6.36 to 2.86?
############################################################################
#response:
- the caption was poorly expressed. The values in brackets for I/sigma, CC1/2, and
completeness refer to the low resolution shell. The low resolution data
contain information about the strength of the data, and the values for the
full data set are sometimes misleading, because the highest resolution shell
may contain a substantial amount of (poor) data. Currently, we include such
seemingly poor data, because they are useful in refinement (cf. Ref. [Si 1],
Karplus and Diederichs)
############################################################################

Reviewer 2 Report

The article presents the application of the very recently developed Jungfrau charge integrating detector, for the detection of electron diffraction in a transmission electron microscope. By using two known aluminosilicates as benchmark samples, the authors demonstrate the high data quality attainable,  showing that structure solution models in which most Al and Si atom types are assigned correctly, can be obtained and refined reliably.

The experiment is certainly novel and the manuscript is tidy and mostly easy to read. I appreciate that the authors give some details of the data processing, because it is still a foggy part of ED.
I am unsure of the significance of this work and of its suitability for the readership of Crystals. The "take home message" I got from reading the article is the qualitative assessment that the Jungfrau detector is suitable for collecting ED data of good quality. It is a proof of concept. However, the uneven level of details of the text indulging in technical aspects of the detector, zeolite structure and properties or in very instrument-specific data collection/processing details, made me at times expect something else. My main concern is that the article should be more focused towards what message the authors want to stand out, keeping the audience in mind. 

General questions to enhance the focus:

GQ1: Single-electron counting hybrid pixel detectors have just started to be applied for ED and they work well and surely cost less than the Jungfrau. Is a charge integrating detector expected to be better for ED than a single electron counter? Or what other characteristics make it worth considering? 

GQ2: I find it a delicate matter to rely on the discrimination of Si vs Al as the major indicator of data quality, so I would highlight also other qualifiers. My concerns are:
(i) one could object that the structure solution program mistakes Na for O which is apparently a bigger mistake than Si/Al. The authors should comment on why this misassignment may be actually less relevant.
(ii) Does ShelxT make use of electron scattering factors?
(iii) Has it been possible discriminate Si/Al in similar experimental conditions but with other detectors? I think it has just not been tried recently. Yet the correct assignment of C/N has been reported. (see: https://landing.eldico-scientific.com/download-application-note-3at )

Specific suggestions on the text:

Lines 1-5: this first sentence is very general and does not match the scope of the article.
Line 27 and elsewhere: maybe the authors could specify "single photon/electron counter"
Line 36: it is not necessary to specify the model of the TEM in the introduction, but if needed I would write the make too.
- The part about zeolites could be made more concise and ordered (alluminosilicates in general; albite; zeolite A), and functional to the results and discussion part. (e.g. no need to describe the topology, but maybe useful to mention how many Na sites are expected).

Line 61: I think the authors mean to say "ab initio", rather than "direct methods". I don't think that SHELXT can be classified as a direct methods program.

Line 79 (and elsewhere): how was the threshold of 10 keV chosen? Were other values also tested?

Lines 80-111 Data acquisition section:

1- what does "SA" "SA20,000" mean? I don't think this is understandable to the average crystallographer. How were these ranges chosen? 

2- what is the beam diameter?

3- it is not mentioned how many data collections were performed, i.e. the reader finds out later that some were at low temperature, and why so?

4- the fact that some data collections were acquired at low or high resolution (why and how) is something that needs to be explained here, otherwise at line 226 it pops out of the blue.

5- Were the unit cells freely refined or constrained?

Lines 113-117: is this level of detail necessary? I have the impression that these calibrations are like the "factory calibrations" for the Jungfrau detector and were reported elsewhere, not done specifically for this experiment. (also it makes me wonder if a calibration with 8 keV photons is really valid for 200 keV electrons). 

Lines 217-218: the sentence is not complete and is repetitive with lines 212-213. 

Lines 232-234 and fig 7: I am not sure that Ti/Ni differentiation by dynamic refinement is to be considered the most recent milestone to overtake. There have been other examples such as C/N, Hg/Pb. Si/Al is undoubtedly a challenge even with no need for comparison.

There are several typos, see highlighted text attached.

Author Response

Comments and Suggestions for Authors

The article presents the application of the very recently developed Jungfrau
charge integrating detector, for the detection of electron diffraction in a
transmission electron microscope. By using two known aluminosilicates as
benchmark samples, the authors demonstrate the high data quality attainable,
showing that structure solution models in which most Al and Si atom types are
assigned correctly, can be obtained and refined reliably.

The experiment is certainly novel and the manuscript is tidy and mostly easy to
read. I appreciate that the authors give some details of the data processing,
because it is still a foggy part of ED. I am unsure of the significance of this
work and of its suitability for the readership of Crystals. The "take home
message" I got from reading the article is the qualitative assessment that the
Jungfrau detector is suitable for collecting ED data of good quality. It is a
proof of concept. However, the uneven level of details of the text indulging in
technical aspects of the detector, zeolite structure and properties or in very
instrument-specific data collection/processing details, made me at times expect
something else. My main concern is that the article should be more focused
towards what message the authors want to stand out, keeping the audience in
mind.

General questions to enhance the focus:

GQ1: Single-electron counting hybrid pixel detectors have just started to be
applied for ED and they work well and surely cost less than the Jungfrau. Is a
charge integrating detector expected to be better for ED than a single electron
counter? Or what other characteristics make it worth considering?

############################################################################
# response: We do not want to rank various detectors. Our main objective is to
report how we use the JUNGFRAU detector for our ED studies, not to claim it is
better than other detectors. We believe that the manuscript reflects our
attitude.

With respect to pricing: Since the Jungfrau is produced using the same
technology as single photon counting hybrid pixel detectors there is no
difference in price. Only in regard to computing infrastructure could it be more
expensive since the data needs more processing. we added a section to the
Introduction about the advantages of charge-integration: "Using a charge
integrating detector has two distinct advantages over photon counting detectors;
at low rates where the tracks from single electrons are visible the deposited
energy per pixel can be used to estimate the entrance point [2] and at high
rates the detector is not sensitive to pulse pileup"
############################################################################

GQ2: I find it a delicate matter to rely on the discrimination of Si vs Al as
the major indicator of data quality, so I would highlight also other qualifiers.
My concerns are: (i) one could object that the structure solution program
mistakes Na for O which is apparently a bigger mistake than Si/Al. The authors
should comment on why this misassignment may be actually less relevant. (ii)
Does ShelxT make use of electron scattering factors? (iii) Has it been possible
discriminate Si/Al in similar experimental conditions but with other detectors?
I think it has just not been tried recently. Yet the correct assignment of C/N
has been reported. (see:
https://landing.eldico-scientific.com/download-application-note-3at )

############################################################################
#response:
ad (i) we added "Note that the sodium atom is not bonded and as higher ADP
values than the framework atoms. Consequently, the local map value is decreased,
which is most likely the reason why it was incorrectly assigned as oxygen." in
subsection 'Benchmark sample Albite'

ad (ii) SHELXT makes use of normalised structure factors. This makes it
independent from the radiation source. We added a reference to Giacovazzo's
textbook 'Fundamentals of Crystallography' in subsection "Structure Solution and
Refinement"

ad (iii) the eldico application note is a nice confirmation of already published
work, like the paracetamol solution we reported in 10.1002/anie.201811318, and
surely also by others. It is possible that others have not looked into the
reproducible discrimination of element types with ED. We believe that this is
even more so a reason to publish our results.
############################################################################

Specific suggestions on the text:

Lines 1-5: this first sentence is very general and does not match the scope of
the article.
############################################################################
# response: in our experience, such a general introduction is important in order
# to reach a broad readership beyond trained crystallographers. Even people with
# training in crystallography do e.g. not always know that differentiation of
# elements is one of the applications in crystallography.
############################################################################

Line 27 and elsewhere: maybe the authors could specify "single
photon/electron counter"
############################################################################
# response: In the detector community the term single photon counter is
# often used regardless of the type of particle detected, but we agree with
# the reviewer that single photon/electron counter is more clear to the reader
# and have adapted the text accordingly.
############################################################################

Line 36: it is not necessary to specify the model of
the TEM in the introduction, but if needed I would write the make too.

######################################################################
# response: we changed this part to "onto a 200 keV transmission electron
# microscope with an LaB_6 electron source,..."
############################################################################

- The part about zeolites could be made more concise and ordered
(aluminosilicates in general; albite; zeolite A), and functional to the
results and discussion part. (e.g. no need to describe the topology, but maybe
useful to mention how many Na sites are expected).

############################################################################
# response: the description of the aluminosilicates has been suggested by the
# co-authors who are specialists in the field. We want to reach a readership
# beyond crystallographers, and consider this depth of description appropriate.
# we addressed the Na sites in the Introduction: " The unit cell is composed of
# eight large alpha-cavities adjacent to one sodalite beta-cage and d4r building
# units that results in a 3D topology. The electronegative framework of zeolite
# A is constructed by 8- 6- and 4-membered rings counterbalanced by 12
# extra-framework Na+ cations placed inside alpha-cage. Sodium sitting in a
# plane of 8-ring controls the aperture size and determines the structure
# properties which have found broad application [18]."
############################################################################

Line 61: I think the authors mean to say "ab initio", rather than "direct
methods". I don't think that SHELXT can be classified as a direct methods
program.
############################################################################
# response: the reviewer seems right. The publication of SHELXT is unclear about
# whether it uses direct methods for structure solution (direct methods improve
# the phases based on a mathematical relationship, e.g. the tangent formula.
# (Giacovazzo, Fundamentals of Crystallography)). Isabel Uson kindly checked
# the source code of SHELXT to confirm that there are direct methods implemented
# in SHELXT (via the -u command line switch). However, using this switch did not
# produce a reasonable solution for one of our data sets, while without '-u',
# the solution just pops out. We removed the mentioning of 'direct methods',
# which did not alter the meaning of the text. In the discussion, we replaced
# 'direct methods' with 'ab initio', as suggested.
############################################################################

Line 79 (and elsewhere): how was the threshold of 10 keV chosen? Were other
values also tested?
############################################################################
# response: The minimum energy that one can reliably recognise as signal
# in a hybrid detector is normally stated at 6 * noise of the system (depending
# on the number of dark counts that are acceptable). For this
# Jungfrau detector operated under these conditions it would be ~3.7 keV.
# In addition we took some extra height to allow for slight pedestal drift
# and include potential low energy X-rays. Rephrased the text on line 79 as:
# During post processing a threshold of 10~keV was applied
# to suppress electronic noise and low energy X-ray background. This value was chosen
# taking into account the measured noise of the detector (0.63 keV) in this configuration and
# allowing some margin for pedestal drift and low energy X-ray background.
############################################################################

Lines 80-111 Data acquisition section:

1- what does "SA" "SA20,000" mean? I don't think this is understandable to the
average crystallographer. How were these ranges chosen?
############################################################################
# response: we replaced this 'TEM' specific slang with a more precise
# description:
#"Albite crystals were small. In their case, the rotation axis was centred
#with a beam diameter of 750 nm, at about 32'000 fold magnification. LTA
#crystals were centred with a beam diameter between 2.3um and
#3um, at 10'500 fold and 8'000 fold magnification."
############################################################################

2- what is the beam diameter?
############################################################################
# response: we added the beam diameters to the method section, cf. previous
# response
############################################################################

3- it is not mentioned how many data collections were performed, i.e. the reader
finds out later that some were at low temperature, and why so?
############################################################################
# response: we added in subsection 'Data Acquisition': " The first few data sets
# of zeolite~A were collected with a cryo holder at -189$^\circ$C. Our
# goniometer stage, however, is optimised for the room temperature holder, and
# data collection was cumbersome, because of drifts during rotation which could
# not be corrected for. We therefore continued data collection with the room
# temperature holder."
############################################################################

4- the fact that some data collections were acquired at low or high resolution
(why and how) is something that needs to be explained here, otherwise at line
226 it pops out of the blue.
############################################################################
# response: we added in subsection 'Instrumentation': "We therefore collected a
# low resolution data set at 780~mm effective detector distance, and a high
# resolution data set at 280~mm effective detector distance in order to cover
# the full resolution range of the samples."
############################################################################

5- Were the unit cells freely refined or constrained?
############################################################################
# response: the unit cell parameters were constrained as indicated by the
# standard uncertainties: where standard uncertainties are given, they are not
# constrained, where none are given, they were constrained to the respective
# Laue group (e.g. angles = 90 degree in Fm-3c) cf. Tables in the SI.
############################################################################

Lines 113-117: is this level of detail necessary? I have the impression that
these calibrations are like the "factory calibrations" for the Jungfrau detector
and were reported elsewhere, not done specifically for this experiment. (also it
makes me wonder if a calibration with 8 keV photons is really valid for 200 keV
electrons).
############################################################################
# response: we consider these details necessary. The JUNGFRAU detector is not a
# commercial product, it is a development project. As stated earlier, we do not
# only want to reach crystallographers with our manuscript.
############################################################################

Lines 217-218: the sentence is not complete and is repetitive with lines
212-213.
############################################################################
# response: thank you for spotting this. we removed the repetitive line.
############################################################################

Lines 232-234 and fig 7: I am not sure that Ti/Ni differentiation by dynamic
refinement is to be considered the most recent milestone to overtake. There have
been other examples such as C/N, Hg/Pb. Si/Al is undoubtedly a challenge even
with no need for comparison.
############################################################################
#response: We are very thankful for this remark. We removed Fig. 7 and the
respective parts of the text. This is consistent with our aim of a neutral
description of the use of the JUNGFRAU detector for ED with this manuscript
rather than a comparison with other detectors.
############################################################################

There are several typos, see highlighted text attached.
############################################################################
#response: thank you, we corrected these typos
############################################################################

Round 2

Reviewer 1 Report

As a reviewer I am satisfied with the changes made and the manuscript in its current form. 

Author Response

Thank you very much for the positive feedback!

Reviewer 2 Report

The manuscript has improved and the authors have taken into account all suggested points. I have only one remark: 

The added text at lines 302-304 mentioning PETS is incorrect.
PETS is a program for data reduction as much as XDS and DIALS, not a program for dynamic refinement. 
Dynamic refinement is implemented in a subprogram of Jana2006. 
Both PETS and Jana2006 are already able to deal with rotation datasets, so it's maybe better to remove these added lines.

Author Response

We meant to make a link between PETS and dynamical refinement. Our attempt came out misleading. We corrected it, and reference some of the recent landmarks for dynamical refinement:

"[...] no attempt to use ionic scattering factors [43]. More sophisticated electron scattering factors than those from [33] are available [43,44]. Furthermore, we processed our data with XDS. Other data reduction programs, like PETS, can be combined with dynamical refinement, e.g. with JANA2006, which can reveal finer structural details than a pure kinematic approach [45–48]."